

# Analysis of the electronic asymmetry of the primary electron donor of photosystem I of *Spirodela oligorrhiza* by photo-CIDNP solid-state NMR

Geertje J. Janssen,[1] Patrick Eschenbach,[2] Patrick Kurle,[3] Bela E. Bode,[4] Johannes Neugebauer,[2] Huub J.M. de Groot,[1] Jörg Matysik,[3]* Alia Alia[1,5]*

[1]Leiden Institute of Chemistry, Leiden University, Einsteinweg 55, P.O Box 9502, 2300 RA Leiden, The Netherlands;
[2]Universität Münster, Organisch-Chemisches Institut and Center for Multiscale Theory and Computation, Corrensstraße 40, D-48149 Münster, Germany
[3]Institut für Analytische Chemie, Universität Leipzig, Linnéstr. 3, D-04189 Leipzig, Germany
[4]EaStCHEM School of Chemistry, Biomedical Sciences research Complex and Centre of Magnetic Resonance, Purdie Building, North Haugh, St Andrews, KY16 9ST, Scotland
[5]Institut für Medizinische Physik und Biophysik, Universität Leipzig, Härtelstr. 16-18, D-04103 Leipzig, Germany

**KEYWORDS:** photosynthesis, photosystem I, radical pair, light-induced electron transfer, MAS NMR.



**ABSTRACT.** The electron donor in photosystem I, the chlorophyll dimer P700, is studied by photo-CIDNP (photochemically induced dynamic nuclear polarization) MAS (magic-angle spinning) NMR on selectively $^{13}$C and uniformly $^{15}$N labeled PSI core preparations (PSI-100) obtained from the aquatic plant duckweed (*Spirodela oligorrhiza*). Light-induced signals originate from the isotope labelled nuclei of the cofactors involved in the spin-correlated radical pair forming upon light excitation. Signals are

assigned to the two donor cofactors (Chl *a* and Chl *a'*) and the two acceptor cofactors (both Chl *a*). Light induced signals originating from both donor and acceptor cofactors demonstrate that electron transfer occurs through both branches of cofactors in the pseudo-$C_2$ symmetric Reaction Center (RC). The experimental results supported by quantum chemical calculations indicate that this functional symmetry occurs in PSI despite similarly sized chemical-shift differences between cofactors of PSI and

the functionally asymmetric special-pair donor of the bacterial RC of *Rhodobacter sphaeroides*. This contributes to converging evidence that local differences in time-averaged electronic ground-state properties, over the donor are of little importance for functional symmetry breaking across photosynthetic RC species.



# INTRODUCTION

In the process of oxygenic photosynthesis, electrons flow from photosystem II (PSII) to photosystem I (PSI), the nomenclature however follows the order of their discovery over time (Emerson and Chalmers, 1958; Govindjee and Rabinowitch, 1960). The X-ray structure of PSI from the prokaryotic system of cyanobacteria Synechococcus elongatus has been solved at 2.5 Å resolution as a trimeric supercomplex (Jordan et al., 2001). In the eukaryotic plant system of Pisum sativum (Pea) the PSI structure has been resolved up to 3.4 Å resolution as a PSI-LHCI complex (Ben-Shem et al., 2003). Cyanobacterial PSI contains 12 subunits with 96 chlorophyll (Chl) cofactors while the plant complex consists of at least 17 subunits harboring over 170 Chls. In cyanobacteria, PSI is mostly observed as a trimer of monomeric PSI cores (Kruip et al., 1994; Fromme et al., 2001), while PSI in plants, red and green algae is monomeric (Scheller et al., 2001; Kouril et al., 2005). Two functional moieties can be distinguished in PSI: the photosystem I core that includes the redox active cofactors, and the peripheral light-harvesting complex (LHCI), which serves to increase the absorption cross section (Schmid et al., 1997; Amunts et al., 2009). While the structural organization of the redox centers is virtually identical in the structures obtained from *Pisum sativum* and Synechococcus elongates (Jordan et al., 2001; Amunts et al., 2007), the LHCI complex shows a high degree of variability in size, subunit composition and number or type of bound pigments. This variation allows each organism to adjust to its specific natural habitat (Croce et al., 2007; Wientjes et al., 2009). The PSI core complex prepared from plants is sometimes also denoted as the PS1-110 particle, referring roughly to the total number (~110) of bound Chls (Mullet et al., 1980) and has a molecular weight of ~300 kDa.

As in PSII and bacterial reaction centers (RC), the cofactors in PSI are symmetrically arranged in two parallel chains relative to a pseudo-$C_2$ symmetry axis perpendicular to the membrane plane in which PSI is embedded in vivo (Figure 1). Like Type-I bacterial RCs, PSI consists of six Chl cofactors, two quinones, and three iron-sulfur [4Fe-3S] clusters ($F_X$, $F_A$, $F_B$) acting as intrinsic electron acceptors. The $F_A$ and $F_B$ clusters operate in series, and are bound to the PsaC subunit. The $F_X$ cluster is located at the interface between the PsaA and PsaB subunits while the accessory Chls ($A_{-1A}$ and $A_{-1B}$), the Chl acceptors ($A_{0A}$ and $A_{0B}$), and the quinone acceptors ($A_{1A}$ and $A_{1B}$) are bound to the PsaA (A-branch) and PsaB (B-branch). In comparison to their PSII quinone counterparts, $A_{1A}$ and $A_{1B}$ in PSI are more tightly associated with the protein backbone and not as readily accessible for chemical reducing agents (Srinivasan et al., 2009). With distances ranging between 15 and 40 Å, the cofactors in the PSI RC are more isolated from the surrounding antenna pigments than the cofactors in PSII and bacterial RC.



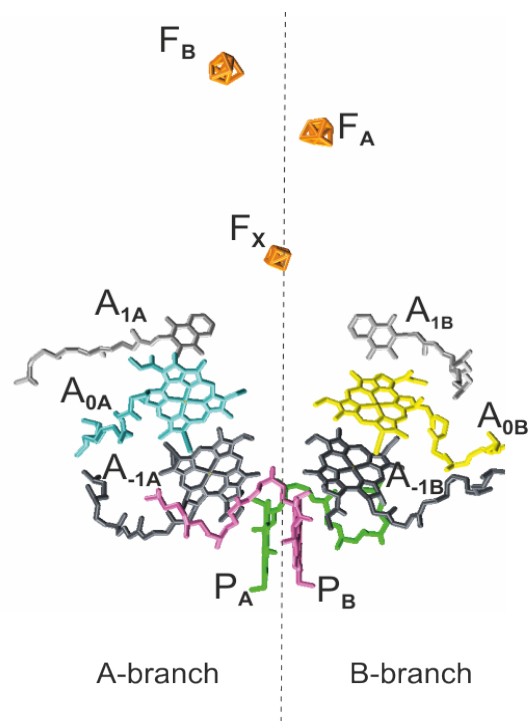

**Figure 1. The arrangement of cofactors in the PS1 RC. Depicted in pink and green are the two central Chls, $P_B$ and $P_A$ of the Chl $a$/Chl $a'$ dimer. Furthermore, the RC contains two accessory Chl $a$ ($A_{-1A}$ and $A_{-1B}$), two donor Chl $a$ ($A_{0B}$ and $A_{0B}$), and two tightly associated phylloquinone's ($A_{1A}$ and $A_{1B}$). Finally, there are three iron-sulphur [4Fe-3S] clusters ($F_X$, $F_A$, $F_B$) which function as terminal intrinsic electron acceptors. Both the A and B branch participate in electron transfer with the relative activity depending mainly on the organism and the reduction conditions. [PDB entry 2WSC (Jordan *et al.*, 2001).**

The electron donor, the heterodimeric P700, is, similarly to the $F_X$ cluster, located at the interface of both branches and consists of one chlorophyll $a$ (Chl a, PB) and one Chl $a'$ (PA), which is the C-$13^2$-epimer of Chl $a$. While $P_A$ forms hydrogen bonds to its protein environment, no hydrogen bonds are found on the PB side (Watanabe et al., 1985). The ratio of the spin-density distribution over the $P_A^{\bullet+}$/$P_B^{\bullet+}$ dimer exhibits significant diversity in between species and conditions (Webber and Lubitz, 2001): Fourier transform infrared and EPR spectroscopic studies on cyanobacterial PSI from *Synechocystis* indicated a ratio of electron spin density distribution in the range of 50:50 to 33:67 in favor of the $P_B$ (Breton et al., 1999). On the other hand, in spinach and *Thermosynechococcus (T.) elongates* ratios in the range of respectively 25:75–20:80 and 15:85 have been estimated (Davis et al., 1993; Käss et al., 2001). Electronic-structure calculations suggested a ratio of 28:72 based on the coordinates taken from the high-resolution X-ray data of *T. elongates* and indicate the hydrogen-bonding of the $P_A$ Chl, the asymmetry in molecular geometry (Chl $a$/Chl $a'$) and minor differences in the protein environment as the main factors influencing the relative spin density distribution over $P_A$ and $P_B$ (Saito et al., 2011).

Calculations making use of the frozen-density embedding technique on the primary electron donor of PSI in *S. elongatus* including a large part of the protein environment resulted in 76% of the spin-density being localized on $P_B$ (Artiukhin et al.,

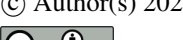



2020). The calculated spin populations were in good agreement with Ref. (Saito et al., 2011) and available experimental data obtained with $^{13}$C photo-CIDNP MAS NMR (Alia et al., 2004).

While the donor in PSII is the strongest oxidizing agent known in living nature, P700 is optimized to provide a strong reducing force, which is required for the formation of NADPH. With a potential of approximately −1.2 V, P700$^{•+}$ is probably the strongest reducing entity found in living systems (Ishikita et al., 2006).

Whereas in Type-II bacterial RCs and PSII, electron transfer (ET) proceeds along only one of the two pseudo-symmetric branches, over the past few years evidence has been accumulated that both branches are active in ET in PSI resulting in a 'bidirectional ET' (see for review: Santabarbara et al., 2010). Since femtosecond optical studies indicated the accessory Chl A$_{-1}$ to be the primary electron donor (Holzwarth et al., 2006; Müller et al., 2010), the structural asymmetry of the P700 Chl *a*/Chl *a*' dimer is no longer a convincing argument against the participation of both branches in ET. A possible reason for the

occurrence of the bidirectional ET in Type-I systems as PSI and RCs of heliobacteria (Thamarath et al., 2012b) but not in Type-II systems as PSII and purple bacterial RC might be that the quinones in Type-II RCs function as a 'two-electron gate', with a mobile quinone on the inactive branch being used as a terminal acceptor (Müh et al., 2012). In Type-I systems, on the other hand, the iron-sulfur clusters act as terminal acceptors, while the quinone serves as an intermediary in electron transfer making bidirectional ET feasible. While consensus on the bidirectional nature of ET in both prokaryotic and eukaryotic PSI

has been reached (Fairclough et al., 2003; Redding et al., 2007), the molecular details controlling the ET pathways are not yet fully elucidated (Berthold et al., 2012). The relative activity of the two branches is in favor of the A-branch, but seems to vary among different organisms ranging from ~3 to 2 in green algae (Holzwarth et al., 2006; Li et al., 2006) to ~3-4 to 1 in cyanobacteria (Ramesh et al., 2004; Dashdorj et al., 2005). The relation between the activity of the ET pathways and the electron (spin) density distribution between the two parts of the donor is not understood. In addition, the reducing conditions

00  of the quinones appear to affect the relative branch activity with, e.g., ET in *Synechococcus lividus* occurring solely along the B branch at low temperature (100 K) and strongly reducing conditions (Poluektov et al., 2005). Hence, the factors inducing the initial asymmetry are not yet understood.

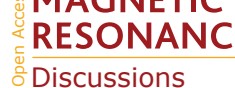

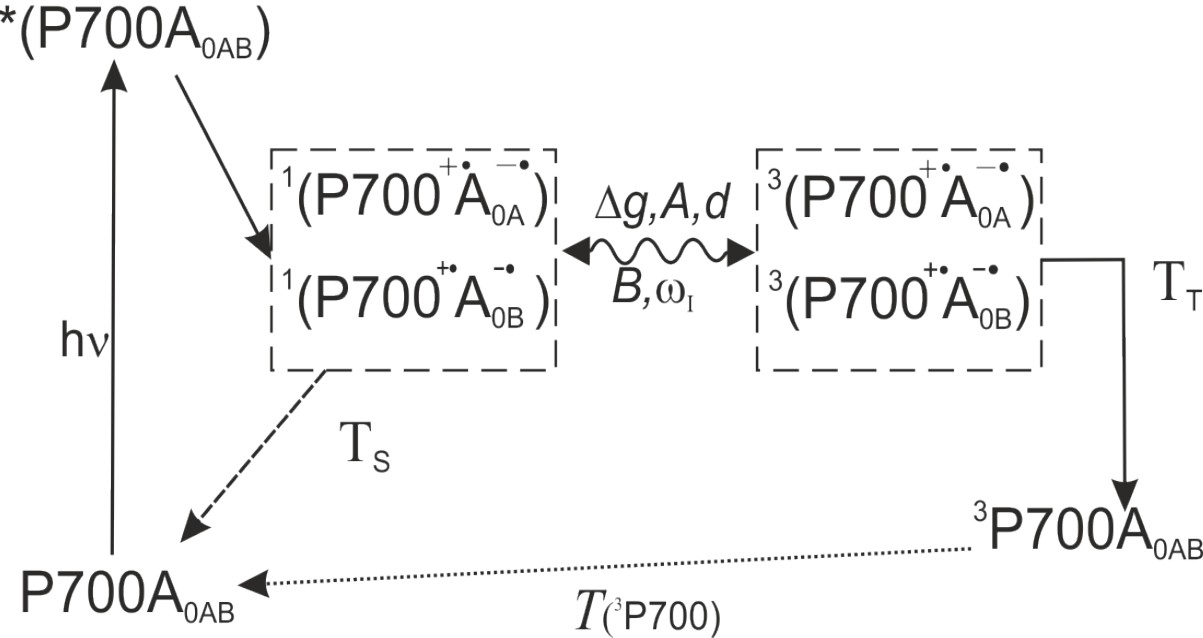

**Scheme 1: Reaction cycle in PSI with reduced $F_X$ acceptor and electron transfer over both branches of cofactors, A and B. After absorption of a photon, electron transfer occurs from the P700 donor to the primary acceptors $A_0$. Upon chemical pre-reduction of $F_X$ the electron transfer becomes cyclic. Due to spin-conservation, the spin-correlated radical pair (SCRP) is formed in a pure singlet state. The SCRP in its singlet state can either recombine to the diamagnetic ground-state or undergo coherent singlet-triplet interconversion to its electronic triplet state. This interconversion relies on the difference of the $g$ values of the two electrons, $\Delta g$, and the hyperfine interaction with magnetic nuclei. According to the radical-pair mechanism (RPM), this influence of the nuclei on the radical pair dynamics leads to spin-sorting and hence to enrichment of nuclear spin states in the two decay channels. In frozen samples, three-spin mixing (TSM) produces nuclear hyperpolarization based on the secular part of the hyperfine interaction, A, the coupling between the two electrons, d, the pseudo-secular hyperfine coupling, B, and the nuclear Zeeman frequency, $\omega_I$. Furthermore, caused by the different kinetics of the two decay channels ($T_S$ vs $T_T$), the differential decay (DD) mechanism for nuclear spin-hyperpolarization occurs. From the triplet state of the SCRP, a molecular donor triplet state is formed which decays with a triplet lifetime, $T(^3P700)$, of ~3 μs (Polm and Brettel, 1998) making the occurrence of the differential relaxation (DR) mechanism unlikely.**

To further investigate the functional symmetry breaking in PSI, we have studied isotope labeled PSI-110 samples form duckweed with photochemically induced dynamic nuclear polarization (photo-CIDNP) magic-angle spinning (MAS) NMR spectroscopy. Photo-CIDNP MAS NMR spectroscopy is an analytical method (for review, see Matysik et al., 2009; Bode et al., 2013) informing on the molecules involved in spin-correlated radical-pairs in both their electronic ground-state (using NMR chemical shift information) and radical-pair state (by photo-CIDNP intensities). The method is based on the solid-state photo-CIDNP effect, discovered in bacterial RCs in 1994 (Zysmilich and McDermott, 1994), occurring in spin-correlated radical pairs (SCRPs) in an immobile matrix upon cyclic ET. The NMR signal is enhanced by up to a factor of 80,000 (Thamarath et al., 2012a). The effect requires a cyclic reaction process that is introduced by pre-reduction of the acceptor site. Scheme 1 shows such a reaction cycle for PSI. In the electronically excited state of the donor (P700*), an electron is transferred to the primary acceptor ($A_0$). The radical pair is formed in the singlet state and undergoes inter-system crossing to





its triplet state. Magnetic coupling to nuclear spins alters the inter-system crossing rates for different nuclear spin states leading to nuclear spin sorting on the singlet and triplet recombination pathways. Although both pathways return to the same product (i.e., the ground state P700-$A_0$), nuclear polarization is generated. The spin-chemical mechanism has been probed by

field-dependent (Thamarath et al., 2012a; Gräsing et al., 2017), time-resolved (Daviso et al., 2009a; Daviso et al., 2010; Sai Sanker Gupta et al., 2014) and preparation-dependent (Matysik et al., 2000a; Daviso et al., 2011) experiments. Nuclear polarization arises from several different mechanisms operating in parallel. The classical radical-pair mechanism (RPM) (Kaptein and Oosterhoff, 1969; Closs and Closs, 1969) relies on spin-sorting and produces transient nuclear polarization in both branches cancelling on arrival of the population of the slower triplet decay channel. In addition, electron-nuclear spin-

dynamics in the radical pair state induces nuclear spin polarization through two solid-state mechanisms called three-spin mixing (TSM, Jeschke, 1997) and differential decay (DD, Polenova and McDermott, 1999) which remains for the period given by the $T_1$ relaxation time. Recently, Sosnovsky et al. (2016; 2019) re-interpreted these coherent solid-state mechanisms in terms of electron-electron-nuclear level-crossings and level anti-crossings. Furthermore, in the differential relaxation (DR) mechanism, also called "cyclic reactions mechanism", the nuclear polarization of the triplet decay channel is quenched by

the paramagnetic molecular triplet state enhancing nuclear relaxation and making cancellation of the RPM polarization incomplete (McDermott et al., 1998).

Various photosynthetic RCs of plants (Alia et al., 2004; Diller et al., 2005; Diller et al., 2007; Janssen et al., 2018), algae (Janssen et al., 2010; Janssen et al., 2012), diatoms (Zill et al., 2017; Zill et al., 2019), purple bacteria (Prakash et al., 2007; Daviso et al., 2009b; Paul et al., 2019), heliobacteria (Thamarath et al., 2012b), green sulfur bacteria (Roy et al., 2008) as

well as flavoproteins (Thamarath et al., 2010; Ding et al., 2019) have been analyzed with the photo-CIDNP MAS NMR method. Previously, the application of $^{13}$C photo-CIDNP MAS NMR has been restricted to the unlabeled and isolated PSI complex due to difficulties in obtaining selective 13C labelling in plants. Based on the data obtained from natural abundance PSI, a first tentative assignment of the light-induced signals involved a single Chl *a* molecule, which is probably the P2 cofactor of the donor P700 (Alia et al., 2004).

In this work, we report the first selective incorporation of $^{13}$C isotope labels in PSI complex from duckweed (*Spirodela*). Backed by $^{15}$N labelling and quantum-chemical calculations, we have explored the photosynthetic machinery of PSI on $^{13}$C and $^{15}$N isotope labelled preparations from duckweed by photo-CIDNP MAS NMR aiming for the details of the electronic structure of the dimeric donor and the question of uni- or bidirectional ET. In addition to continuous illumination with white light, $^{13}$C photo-CIDNP MAS NMR was induced by a 532-nm nanosecond flash laser.




## MATERIALS AND METHODS

### Photosystem I Particle Preparation

Duckweed plants were grown under aseptic conditions on half-strength Hunter's medium (Posner 1967) under continuous light (20 µEm$^{-2}$s$^{-1}$) at 25 °C. The medium was continuously bubbled with sterile air containing 5% $CO_2$. For selective $^{13}$C
labeling plants were exposed to δ-aminolevulinic acid (ALA, purchased from Cambridge Isotope Laboratories), isotopically $^{13}$C labeled at carbon position 4 (4-ALA) to a final concentration of 1.4 mM in half-strength Hunter's medium at pH 4.8. After 7 days, plants were harvested and used directly for sample preparation or frozen in liquid nitrogen and stored at -80 °C until use. The PSI complex containing ~110 Chl/P700 (PS1-110 particles) was prepared according the method described by Alia et al. (2004).

### Determination of the $^{15}$N, $^{13}$C-label incorporation

Chl *a* were extracted from plants grown in ALA supplemented half-strength Hunter's medium (labeled sample) and from unlabeled plants (reference sample), according to the procedure of Moran and Porath (1980). Plants were homogenized in MeOH. The methanolic solution was centrifuged for 5 min at 300×*g*. The green supernatant was separated and dried under a gentle stream of $N_2$. The sample was re-suspended in acetone, loaded on a cellulose column and pure Chl *a* fractions were
eluted with petroleum ether/acetone (7/3 *v/v*). The solvent was evaporated under a $N_2$ flow and the pure Chl *a* was stored at -20 °C in a dry nitrogen atmosphere. Label incorporation has been determined by mass spectrometry to be about 75% for each particular carbon position of the 4-ALA isotope label pattern. For details, see SI.

### Photo-CIDNP MAS NMR experiments

The NMR experiments were performed by using DMX/AV-100, -200, -300 and -400 NMR spectrometers (Bruker GmbH,
Karlsruhe, Germany). The samples were loaded into optically transparent 4-mm sapphire rotors. The PSI samples were reduced by the addition of an aqueous solution of 10 mM sodium dithionite solution prepared in 40 mM glycine buffer (*p*H 9.5) in an oxygen free atmosphere. Immediately following the reduction, slow freezing of the sample was performed directly in the NMR probe inside the magnet under continuous illumination with white light. All spectra have been obtained at a sample temperature of 235 K and with a spinning frequency of 8 kHz.
The spectra were collected with a spin-echo pulse sequence with a phase cycle of (π/2) pulses under two-pulse phase modulation (TPPM) carbon-proton decoupling (Bennett et al., 1995). Photo-CIDNP MAS NMR spectra have been obtained using continuous illumination with a 1000 Watt Xenon arc lamp (Matysik et al., 2000b). The number of scans was 20 k, unless stated differently. The fitting of the collected spectra was performed using Igor Pro 6.01, based on the relative intensity of the signals, the electron spin density was calculated for the nitrogen assigned to the donor.

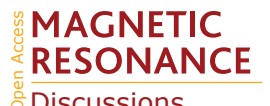

A pulsed nanosecond-flash laser provides sufficient radiation intensity for time-resolved photo-CIDNP MAS NMR studies and does not decrease the time-resolution that can be obtained in NMR experiments. The laser is operating with a repetition rate between 1 and 10 Hz. Using 1064-nm flashes of a Nd:YAG laser (SpectraPhysics Quanta-Ray INDI 40-10, Irvine, CA, USA), upon frequency-doubling with a second harmonic generator (SHG), 532-nm laser flashes with pulse length of 6–8 ns and an energy between 20 and 150 mJ are produced.

**Quantum-chemical calculations**

The structural models employed in our calculations were extracted from the crystal structure of Photosystem I in plants (PDB entry 2WSC (Amunts et al., 2010)), provided by the Protein Data Bank (Berman et al., 2000). Two different types of molecular models were considered: the "iso" models correspond to the isolated co-factors extracted from the crystal structure. The binding pocket models abbreviated by "r32" and "r34" were created by specifying radii of 3.2 and 3.4 Å

around each atom of the co-factor of interest. All surrounding co-factors, water molecules, and amino acid residues with at least one atom within these radii were included explicitly into these models. For geometry optimizations, the DFTB3 (Gaus et al., 2012) method within the AMS-DFTB module from the ADF 2019 package (Amsterdam; Velde et al., 2001) was used. The "Third-Order Parametrization for Organic and Biological Systems" (3ob) (Gaus et al., 2013; Kubillus et al., 2015) parameters from the corresponding Slater–Koster file were used. The optimizations were performed as a sequence of several

00    steps that partly optimize the protein structure. For further details on the model setup and geometry optimization see Sections 2.1 and 2.2 of the Supporting Information. Graphical examples of the generated structures can be found in Section 5 of the Supporting Information.

NMR calculations of the binding pocket models "r32" or "r34" were carried out within a subsystem DFT approach (Jacob and Visscher, 2006) using the TZP (van Lenthe and Baerends, 2003) basis set and the PW91 (Perdew et al., 1992; Perdew and Wang, 1991) XC functional with the conjoint (Lee et al., 1991) kinetic-energy functional PW91k (Lembarki and

05    Chermette, 1994). $^{15}$N chemical shifts were calculated with respect to the ammonia shieldings, while $^{13}$C chemical shifts were calculated with respect to chlorotrimethylsilane. Ring current effects of other subsystems were considered by calculating nuclear independent chemical shifts (NICS) following Jacob and Visscher (2006). For further details on the NMR calculations see Sections 2.3 - 2.5 of the Supporting Information.

10    **RESULTS AND DISCUSSION**

$^{15}$N **photo-CIDNP MAS NMR -** Figure 2 shows $^{15}$N MAS NMR spectra of uniformly $^{15}$N labeled PSI-110 particles of duckweed obtained under continuous illumination with white light at magnetic field strengths of (A) 2.35, (B) 4.7, (C) 7.1 and (D) 9.4 Tesla. At higher fields, the signal of the amide backbone nitrogens of the protein becomes clearly visible at about 125 ppm as a broad peak. In addition, sharp light-induced emissive (negative) signals were observed originating from the



Chl *a* and Chl *a*' cofactors involved in formation of a SCRP. All light-induced signals are emissive at all magnetic fields investigated, and the absolute intensity increases with the magnetic field strength. Previous numerical simulations suggest that the matching conditions of the enhancement mechanisms are best met at 9.4 Tesla (i.e., 400 MHz [1]H frequency) leading to maximum signal enhancement (Roy et al., 2007). The three emissive [15]N signals appear at 254 (strong, with shoulder at 250), 210 (very strong, with weak shoulder at 207) and 188 (medium) ppm. The signals are in good agreement with previous [15]N photo-CIDNP MAS NMR data of PSI from duckweed and spinach obtained at 4.7 T (Janssen et al., 2012) and can be conveniently assigned to a Chl *a* cofactor having signals at 247.0, 189.4, 206.5 and 186.6 ppm in solution NMR for N-IV, N-III, N-II and N-I, respectively (Boxer et al., 1974). The strongest signal belongs to a single N-II, and the second strongest originates from the N-IV nitrogen. Whether the third signal occurring at 188 ppm originates from either N-I or N-III is not clear. Since shoulders and asymmetries occur, it appears that the signals originate from multiple cofactors. Emissive signals can arise from either donor or acceptor cofactors, therefore the sign cannot indicate the site of origin of signals. Since a chemical shift assignment would allow to recognize whether the signals originate from donor or acceptor, and if from the donor, whether from Chl *a* or Chl *a*', we performed quantum-chemical calculations to estimate the chemicals shifts of the different cofactors.

**MAGNETIC RESONANCE** Discussions — Open Access

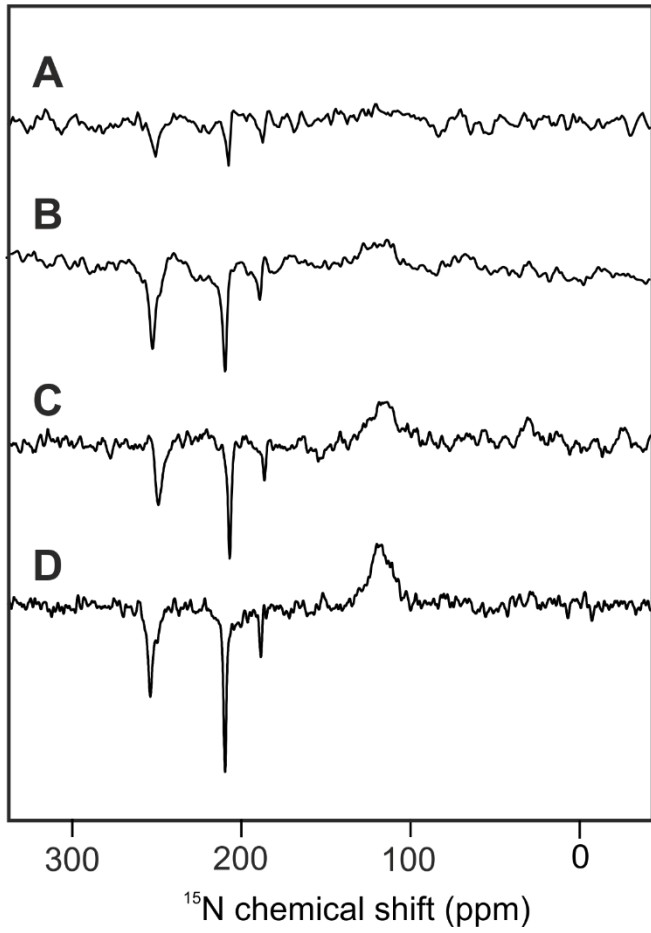

**Figure 2. $^{15}$N photo-CIDNP MAS NMR spectra obtained from the same sample of uniformly 15N labeled PSI-110 particles of duckweed measured at magnetic field strengths of (A) 2.35 T, (B) 4.7 T, (C) 7.1 T and (D) 9.4 T. All spectra have been obtained with a MAS frequency of 8 kHz, a temperature of 235 K, a cycle delay of 4 s and an illuminance of 320 kLux provided by a white Xenon lamp. The number of scans was kept constant.**

The calculated $^{15}$N chemical shifts of the two donor cofactors $P_A$ and $P_B$ as well as the two acceptor cofactors $A_{0A}$ and $A_{0B}$

are shown in Tables S2.1 and S2.2, respectively. The general chemical-shift pattern is well reproduced by the calculations, however, assignment of resonances to specific cofactors is not possible. A possible source for deviations from the calculated NMR shifts arises from the use of the static crystal structure rather than averaging over conformations accessible during the protein dynamics. This could be assessed by performing a short molecular dynamics simulation and calculating NMR shifts for an ensemble of structures, which is, however, beyond the scope of this work. Also, the inclusion of a sphere of protein

environment of 3.2 and 3.4 Å ("r32" and "r34" in Tables S2.1 and S2.2 and Figures S3.1 to S3.4) does not allow for a conclusive assignment. Although there is a significant environmental effect predicted, the strongest experimentally observed signal, N-II at 210 ppm, might be tentatively assigned either to $P_A$ or to $A_{0B}$ having very similar values. The experimental signal at 254 ppm (N-IV) might be tentatively assigned to the donor cofactors having a significantly larger chemical shifts





than the acceptor signals. The two epimeric cofactors forming the donor cannot be distinguished on the basis of calculated chemical shifts. Hence, PSI, having four similar cofactors which might be involved into the formation of the SCRP, does not allow for straightforward $^{15}$N chemical shift assignment.

**$^{13}$C photo-CIDNP MAS NMR -** To further characterize the individual cofactors of PSI involved in SCRP, we next performed $^{13}$C photo-CIDNP MAS NMR on selectively $^{13}$C labelled PSI. Previously, $^{13}$C photo-CIDNP MAS NMR studies on plant PSI have been restricted to experiments on unlabeled preparations due to the difficulty to incorporate selective $^{13}$C isotope labels into plant RCs. In the present study, we succeeded to incorporate selectively 4-ALA in PSI from duckweed with isotope enrichment of 75% for each particular carbon position of the 4-ALA isotope label pattern (Figure 3A).

The $^{13}$C NMR spectra in Figure 3B are obtained from 4-ALA labelled PSI-110 preparations at a magnetic field strength of 4.7 T ($^{1}$H frequency of 200 MHz, spectra a) and 9.4 T (400 MHz, spectra b) obtained under continuous light or in the dark. The spectra under illumination show several light-induced signals (shown in red) which are not observable in the dark (shown in black). The light-induced signature, however, is very different at the two magnetic fields. While the light-induced signals obtained at 4.7 T are entirely enhanced absorptive, at 9.4 T most of the signals appear emissive. Significant magnetic-field effects have been observed for RCs of heliobacteria (Thamarath et al., 2012b) and purple bacteria (Thamarath et al., 2012a), and a similar dramatic sign change has been observed very recently in $^{13}$C MAS NMR spectra of natural abundance PSII preparations from the diatom *Phaeodactylum tricornutum* (Zill et al., 2019). For unlabeled PSI-110 preparations of spinach (Alia et al., 2004), an entirely emissive envelope has been observed at 400 MHz. In the present study, however, some signals appear to turn positive suggesting that $^{13}$C isotope labeling indeed has some influence on the solid-state photo-CIDNP mechanisms. The entirely emissive envelope observed at higher fields suggests the absence of contributions by the DR mechanism, which is reasonable due to the presence of carotenoids, implying that the solid-state photo-CIDNP effect relies on DD and TSM mechanisms. Since the ratio between TSM and DD is field-dependent (Jeschke and Matysik, 2003) our data suggest that the TSM, expected to cause emissive signals (Prakash et al., 2005), contributes more strongly, while the DD decays at higher fields.

A more detailed view on the light-induced signals is provided in Figure 3C. The chemical shifts of the observed lines in the spectra are listed in Table 1. Careful examination of the spectra show that the emissive signals observed at higher field (Spectrum 3B, labelled in red) are cancelled at lower field, while the positive signals are visible in both spectra (labelled in black). Therefore, it appears that the signals belong to two different sets. One might assume that one set originates from the donor and the other from the acceptor cofactors of PSI. The occurrence of the emissive signal at 51.2 ppm at 9.4 T originating from a C-17, the only aliphatic labelled position, is due to spin-diffusion, i.e. polarization transfer from near-by $^{13}$C -labelled aromatic carbons. The cancellation of this signal at 4.7 T implies that at that field also the near-by aromatic carbons do not obtain enhancement.

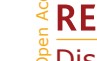

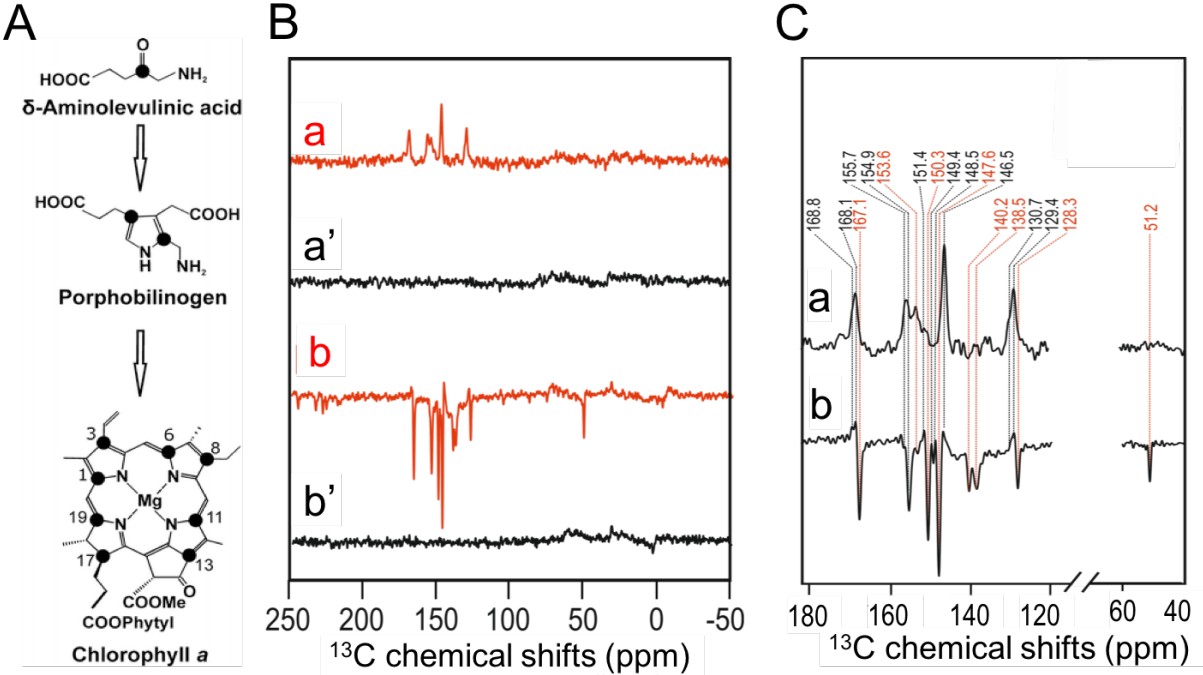

**Figure 3. (A) Incorporation of [4-$^{13}$C]-ALA into cofactors (e.g. Chl *a)* of PS1-110 of duckweed. The black dots indicate positions of $^{13}$C isotopes. (B) $^{13}$C photo-CIDNP MAS NMR spectra obtained under continuous illumination (red) of 4-ALA labeled PS1-110 particles of duckweed at magnetic field strength of 4.7 T (a), and 9.4 T (b). Spectra a' and b' depicted in black originate from the corresponding experiments obtained in the dark. (C) Zoomed view of $^{13}$C photo-CIDNP MAS NMR spectra of 4-ALA labeled PS1-110 particles of duckweed obtained at 4.7 T (a) and 9.4 T (b). Assigned signals are visualized by the dashed lines, emissive signals are colored in red.**

All light-induced signals can be assigned conveniently to respective $^{13}$C labelled carbon positions of cofactors (indicated in red in Table 1). Since there is no light-induced signal which requires assignment to a non-labelled position, we assume that all observed signals originate from $^{13}$C-labelled carbons. This assumption is reasonable considering the enrichment factor of

85 75%. For several of the labelled $^{13}$C positions, multiple signals are observed which support the conclusion from the $^{15}$N data that several cofactors are observed. Three signals can be assigned to the carbons C-19 and C-13. For position C-11, four signals can be resolved. This observation strongly suggests that all four cofactors experience signal enhancement implying that all four cofactors are involved in the spin-correlated radical pair and confirming that both electron transfer pathways are active.



**Table 1.** **¹³C chemical shifts of the photo-CIDNP signals obtained at 4.7 and 9.4 Tesla in comparison to literature data. Assignments obtained from 4-ALA labelled samples are labelled in red.**

| Carbon number (4-ALA label) | ¹³C chemical shifts (ppm) | | | | |
|---|---|---|---|---|---|
| | Chl *a* | Tentative assignments | | | |
| | | n.a. plant | 4-ALA cyanobact., 4.7 T | 4-ALA plant | |
| | | | | 4.7 T | 9.4 T |
| | $\delta^a$ | $\delta^b$ | $\delta^c$ | $\delta^d$ | $\delta^d$ |
| 19 | 170.0 | 167.1 E | 166.9 E | 168.5 A | 168.8 A, 168.1 A, 167.1 E |
| 14 | 162.0 | 160.4 E | | | |
| 1 6 | 155.9 154.4 | 158.4 E | 154.8 E | ≈ 155 A 153.6 A | 154.9 E 153.6 E |
| 16 | 154.0 | 152.6 E | | | |
| 4 | 150.7 | 149.9 E | | | |
| 11 9 | 147.2 | 147.2 E | 149.8 E 147.6 E | 150.3 A | 151.4 A, 150.3 E, 149.4 E, 147.6 E |



| | | | | | |
|---|---|---|---|---|---|
| 8 | 146.2 | 144.2 E | 144.2 E | 146.5 A | 146.5 A |
| 3 | 138.0 | 138.6 E | 138.6 E | | 140.2 E, 138.5 E |
| 2 | 136.1 | ≈ 136 E | | | |
| 12 | 134.0 | | | | |
| 7 | 133.4 | ≈ 132 E | | | |
| 13 | 126.2 | | | 130.7 A, 129.4 A | 130.7 A, 129.4 A, 128.3 E |
| 10 | 108.2 | 105.4 E | | | |
| 15 | 102.8 | | | | |
| 5 | 98.1 | | | | |
| 20 | 93.3 | | | | |
| 17 | 51.4 | | 53.9 E | | 51.2 E |

[a] **Boender (1995), data experimentally obtained from solid aggregates of Chl *a*.**

[b] **Alia et al. (2004), data experimentally obtained from isolated PSI particles from spinach.**

[c] **Janssen et al. (2010), data experimentally obtained from 4-ALA labelled whole cells of *Synechocystis* containing both, PSI and PSII.**

[d] **This work. Data experimentally obtained from 4-ALA labelled isolated PSI from duckweed.**

**A = absorptive (positive), E = emissive (negative) signal intensity, n.a. = natural abundant isotope distribution.**

**Red numbers = $^{13}$C isotopically labelled in the 4-ALA pattern (Fig. 3).**



To explore whether the assignment can be improved by attribution to individual cofactors, occurring from the aromatic $^{13}$C carbons, quantum chemical calculations have been performed for the bare cofactors and including surrounding amino acids up to a shell of 3.4 Å (Tables S2.3 and S2.4). For C-11, four experimental values of 151.4, 150.3, 149.4 and 147.6 ppm have been observed. The calculated shifts for C-11 span a similar range of about 5 ppm. In general, the calculated values for the other carbon positions confirm this finding: The differences between the four cofactors are in the range of about less than 5 ppm in PSI. The differences in chemical shifts between the two BChl cofactors of the Special pair in *R. sphaeroides* were found to be slightly larger in previous studies (Schulten et al., 2002; Daviso et al., 2009b; Sai Sankar Gupta et al., 2014). However, this relatively limited difference in chemical-shift asymmetry cannot explain the fundamentally different functional asymmetry in the bacterial RC. Since chemical shift refers essentially to time-averaged electronic ground state properties, it is tempting to conclude that the different behavior of donor dimers is encoded in the dynamic structure. This is corroborated by studies of the functional symmetry breaking involving the special pair in bacterial RCs, which is thought to originate from specific long living cooperative modes for semiclassical coherent mixing of charge transfer character into the electronically excited state from which the electron is transferred (Thamarath et al., 2012a) and local differences in molecular dynamics affecting the electron-phonon coupling (e.g.: Novoderezhkin et al., 2004; Wawrzyniak et al., 2011).



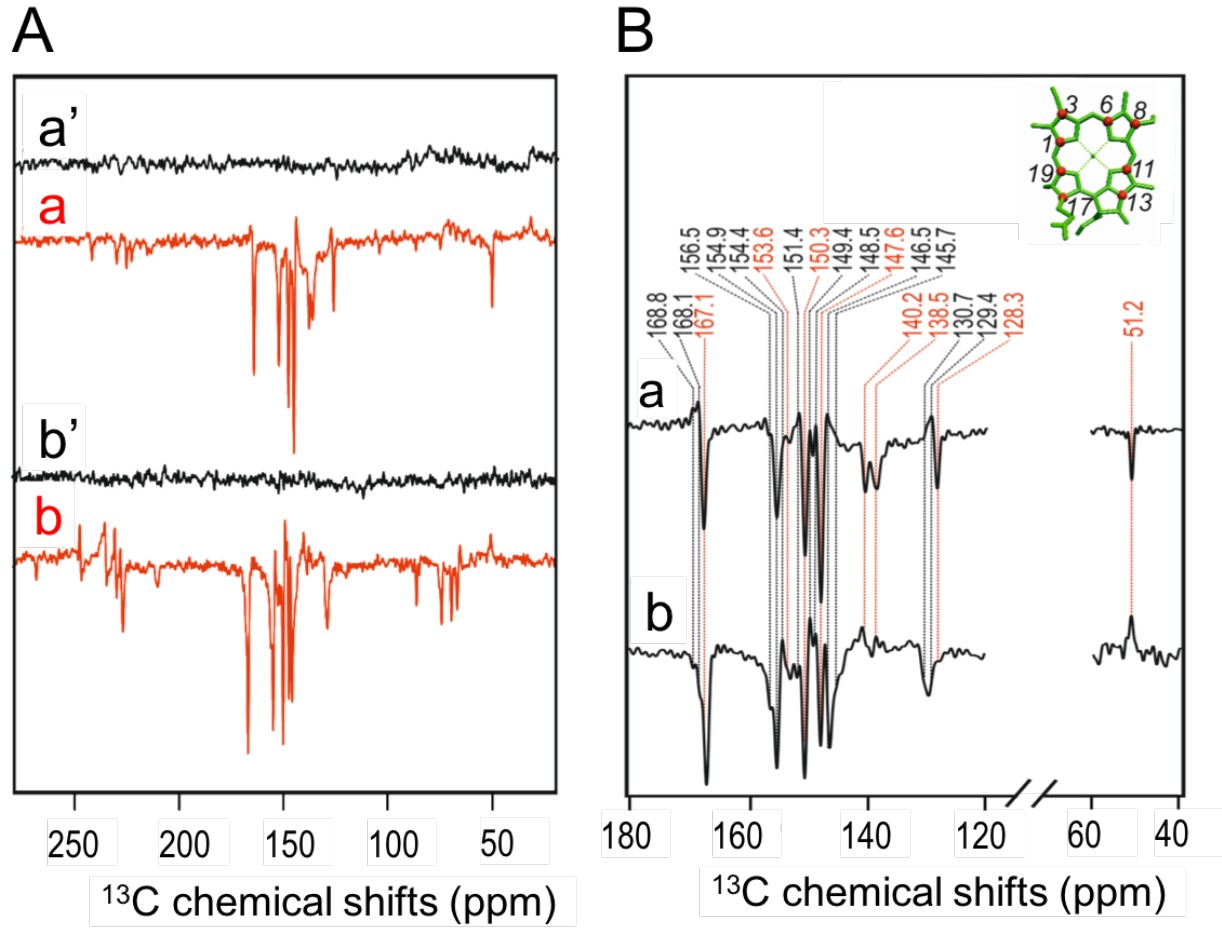

**Figure 4. (A) $^{13}C$ photo-CIDNP MAS NMR spectra of 4-ALA labeled PS1-110 particles of duckweed obtained at 9.4 T under continuous illumination (a) and under nanosecond laser flashes with zero delay (b). Spectra a' and b' depicted in black originate from the corresponding experiments obtained in the dark. (B) Zoomed region of $^{13}C$ photo-CIDNP MAS NMR spectra of 4-ALA labeled PS1-110 particles of duckweed obtained at 9.4 T under continuous illumination (a) and under nanosecond laser flashes with zero delay (b). Assigned signals are visualized by the dashed lines, emissive signals are colored in red.**

Figure 4 compares the $^{13}C$ photo-CIDNP MAS NMR spectrum induced by white light upon continuous illumination (spectrum 4Aa) with that induced by a 532-nm nanosecond flash laser (spectrum 4Ab). The magnified view of both light-induced spectra is shown in Fig. 4B. Remarkably, in the spectrum obtained by the laser-flash experiment, several signals changed their sign. Enhanced absorptive signals turned emissive for the peaks at 168.8, 168.1 (both are assigned to C-19) and 129.4 ppm (C-13). On the other hand, several emissive signals become enhanced absorptive at 140.2, 138.5 ppm (both arise from a C-3), and at 51.2 ppm (C-17). The different intensity patterns are due to differences in the enhancement mechanisms: Under steady-state illumination, the solid-state mechanisms TSM and DD produce the hyperpolarization which rely on anisotropic hyperfine interactions. In time-resolved experiments, the first detectable signals mainly refer to the





singlet branch of the RPM and are based on isotropic hyperfine interactions. Equilibration of the polarization between the carbons by spin-diffusion occurs on slower time-scale (Daviso et al., 2009a).

## CONCLUSIONS

There is experimental evidence that both cofactors of the donor ($P_A$ and $P_B$) as well as both potential acceptor cofactors ($A_{0A}$ and $A_{0B}$) carry electron spin density of the spin-correlated radical pair. This confirms that both electron pathways in PSI of duckweed are active and that the electron transfer occurs symmetrically. In addition, the time-averaged ground state electron density as measured by the chemical shift varies to a similar extend as in the functionally asymmetric special pair of RCs of *R. sphaeroides*. Our study contributes to converging and convincing evidence that the breaking of functional symmetry is not primarily due to local variation in time-averaged electronic ground-state properties at the donor site, but, for instance, local and global electronic excited state properties in conjunction with molecular dynamics.

35

40

**MAGNETIC RESONANCE**
Open Access Discussions

## ASSOCIATED CONTENT

**Supporting Information.** (1) Determination of the isotope incorporation. (2) Computational Details. (3) Chemical shifts calculated by quantum-chemical methods. (4) Effect of the Protein Environment on the Calculated NMR Shifts. (5) Graphical Examples of Structures.

## AUTHOR INFORMATION

### Corresponding Authors

*A. Alia, Leiden Institute of Chemistry, Leiden University, Einsteinweg 55, P.O Box 9502, 2300 RA Leiden, The Netherlands, e-mail: a.alia@chem.leidenuniv.nl. *J. Matysik, Institut für Analytische Chemie, Universität Leipzig, Linnéstr. 3, 04103 Leipzig, Germany, e-mail: joerg.matysik@uni-leipzig.de, fax: +49 341 9736115.

### Author Contributions

H.J.M.d.G., J.M. and A.A. designed the research, G.J.J. and A.A. prepared the samples. G.J.J. measured the NMR spectra, P.E., B.E.B. and J.N. provided the quantum chemical calculations, G.J.J., P.K., B.E.B., J.M. and A.A. interpreted the data. The manuscript was written through contributions of all authors. All authors have given approval to the final version of the manuscript.

### Funding Sources

The work has been supported generously by the Dutch Science Organization (NWO, grant 818.02.019), the Deutsche Forschungsgemeinschaft (DFG, MA 497/2-1 and 11-1) as well as an Alexander-von-Humboldt grant and a Marie-Curie grant to B.E.B.

### Notes

The authors declare no competing financial interests.



## ACKNOWLEDGMENT

The authors would like to thank to Dr. K.B. Sai Sankar Gupta, F. Lefeber and K. Erkelens for their kind help and Dr. Peter Gast and Dr. Hans van Gorkom (Univ. Leiden) for stimulating discussions.

## ABBREVIATIONS

70 ALA, δ-aminolevulinic acid; Chl, chlorophyll; DD, differential decay; DFT, density functional theory; DR, differential relaxation; ET, electron transfer; light-harvesting complex, LHCI; MAS, magic-angle spinning; NMR, nuclear magnetic resonance; P, primary donor; Phe, pheophytin; photo-CIDNP, photochemically induced dynamic nuclear polarization; PS, photosystem, RC, reaction center; RPM, radical pair mechanism; SCRP, spin-correlated radical pair; TSM, three-spin mixing.

75



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
