# Peer review of "Analysis of the electronic structure of the primary electron donor of photosystem I of *Spirodela oligorrhiza* by photo-CIDNP solid-state NMR"

_Magnetic Resonance, 2020_

## Referee Comment (RC1) · Peter Hore (Referee) · 2 Oct 2020

**Analysis of the electronic asymmetry of the primary electron donor of photosystem I of *Spirodela oligorrhiza* by photo-CIDNP solid-state NMR**

Janssen, Eschenbach, Kurle, Bode, Neugebauer, de Groot, Matysik, Alia

This manuscript uses the technique of photo-CIDNP MAS NMR to show that $^{13}$C chemical shift differences between cofactors do not correlate well with the symmetry/asymmetry of electron transport in photosystem I from duckweed. This result provides indirect support for the hypothesis that differences in molecular dynamics and electronic excited state properties are responsible for breaking the functional symmetry of the reaction centre.

**Specific comments**

1.  The Materials & Methods section gives rather little information on the procedures the authors have developed to incorporate $^{13}$C-labels into PSI particles. For example what does "exposed" in "plants were exposed to δ-aminolevulinic acid" (line 160, page 6) actually mean? Given that this is the _first_ time that anyone has managed selective incorporation of $^{13}$C isotope labels into PSI from duckweed, I think there should be a little more detail on how this was achieved so that others will be able to do similar experiments in future.
2.  We are told (page 11) that it was not possible to assign the $^{15}$N resonances to specific cofactors on the basis of the chemical shifts. Is there any information in the relative CIDNP enhancements (Fig. 2) that could help in this regard?
3.  On page 12, it is not clear why the CIDNP enhancement of C17 must be the result of spin-diffusion. Does it have a negligible hyperfine interaction? Are there no other labelled aliphatic carbons that might receive polarization by spin-diffusion?
4.  The heading of columns 3-6 of Table 1 is "Tentative assignments". Are these the same tentative assignments that "strongly suggest" (page 13) that all four cofactors are involved in the spin-correlated radical pair and therefore that both electron transfer pathways are active? Since this is one of the main conclusions of the paper, I think there should be a bit more discussion of how it was reached.
5.  I am afraid that Table 1 is a mess. Some horizontal lines to separate the entries for the different carbons would make it much more readable. For example, at the bottom of page 14 in the right-most column, there appears to be 1 assignment (coloured red) for C11 (coloured red in the left-most column) and three red assignments for C9 (coloured black in the left-most column). Given that red means 4-ALA labelled and black means literature values, either some of the colours are wrong or all four of the 9.4 T assignments actually belong to C11 and none of them to C9.
6.  If all four entries at the bottom right of Table 1 on page 14 are in fact for C11, can anything be learnt from the fact that one is absorptive and the other three emissive?
7.  Page 18: "This confirms that both electron pathways in PSI of duckweed are active and that the electron transfer occurs symmetrically". Symmetrically suggests 50:50 along the two branches. If this is the intended meaning, then I'm not sure where this ratio comes from. Or is "symmetrically" being used rather loosely to mean something like "not exclusively by one branch".

**Minor comments**

Lines 47, 73, 75 : elongates → elongatus?

Line 69: PB $\rightarrow$ P$_B$

Page 18, line 337: extend $\rightarrow$ extent

---

## Author Comment (AC1) · 9 Oct 2020

**POINT-BY-POINT REPLY TO REVIEWERS´ COMMENTS**

Analysis of the electronic asymmetry of the primary electron donor of photosystem I of *Spirodela oligorrhiza* by photo-CIDNP solid-state NMR
Janssen, Eschenbach, Kurle, Bode, Neugebauer, de Groot, Matysik, Alia

**REVIEWER PETER HORE**

This manuscript uses the technique of photo-CIDNP MAS NMR to show that $^{13}$C chemical shift differences between cofactors do not correlate well with the symmetry/asymmetry of electron transport in photosystem I from duckweed. This result provides indirect support for the hypothesis that differences in molecular dynamics and electronic excited state properties are responsible for breaking the functional symmetry of the reaction centre.
That is truly the essence of the manuscript.

Specific comments
1. The Materials & Methods section gives rather little information on the procedures the authors have developed to incorporate $^{13}$C-labels into PSI particles. For example what does "exposed" in "plants were exposed to δ-aminolevulinic acid" (line 160, page 6) actually mean? Given that this is the first time that anyone has managed selective incorporation of $^{13}$C isotope labels into PSI from duckweed, I think there should be a little more detail on how this was achieved so that others will be able to do similar experiments in future.
It is indeed the first time that $^{13}$C incorporation into duckweed is reported. Following the suggestion of the reviewer, we extended that section and added more information (line 158 ff).

2. We are told (page 11) that it was not possible to assign the $^{15}$N resonances to specific cofactors on the basis of the chemical shifts. Is there any information in the relative CIDNP enhancements (Fig. 2) that could help in this regard?
In this case, the signal envelope is formed by contributions from four Chl cofactors having similar chemical shifts. It might be that the envelope is composed from enhanced absorptive and emissive contributions, too. An dis-entanglement appears not possible at the present stage of simple one-dimensional $^{15}$N-NMR.

3. On page 12, it is not clear why the CIDNP enhancement of C17 must be the result of spin-diffusion. Does it have a negligible hyperfine interaction? Are there no other labelled aliphatic carbons that might receive polarization by spin-diffusion?
Here the reviewer detected that an explanation needs to be added: signals from C-17, as aliphatic carbon, do not profit from the solid-state photo-CIDNP effect. The TSM, for example, requires hyperfine anisotropy and is therefore related to the electron-spin densities in $p_z$-orbitals. Therefore, if C-17 appears enhanced, the nucleus must have been receiving polarization from a near-by aromatic carbon as C-19. We added a statement accordingly (line 271).

4. The heading of columns 3-6 of Table 1 is "Tentative assignments". Are these the same tentative assignments that "strongly suggest" (page 13) that all four cofactors are involved in the spin-correlated radical pair and therefore that both electron transfer pathways are active? Since this is one of the main conclusions of the paper, I think there should be a bit more discussion of how it was reached.
We agree with the reviewer. We stressed the basis of our assignment: selectively only $^{13}$C labelled carbons appear allowing for a consistent set of assignments. Therefore, we removed the word "tentatively" from Table 1. We also stressed that signals from C-13 and C-19, which are very isolated, appear three times, also suggesting that not a single branch is active as in photosystem II.

5. I am afraid that Table 1 is a mess. Some horizontal lines to separate the entries for the different carbons would make it much more readable. For example, at the bottom of page 14 in the right- most column, there appears to be 1 assignment (coloured red) for C11 (coloured red in the left- most column) and three red assignments for C9 (coloured black in the left-most column). Given that red means 4-ALA labelled and black means literature values, either some of the colours are wrong or all four of the 9.4 T assignments actually belong to C11 and none of them to C9.

We agree and followed the suggestions: we added horizontal lines, added a statement that carbon C-9 and C-11 were not be possible top separate in the reference experiment. Therefore, C-9 appears now in a line on top of C-11.

6. If all four entries at the bottom right of Table 1 on page 14 are in fact for C11, can anything be learnt from the fact that one is absorptive and the other three emissive?

That is a nice and exciting question. A straightforward answer might be: A radical pair occurring one branch has different distance to carotenoids than the pair on the other branch. Therefore one pair (undergoing TSM and DD evolution) shows only emissive signals (as RCs of *Rhodobacter sphaeroides* WT), while the other radical pair (also affected by the DR mechanism) shows enhanced absorptive signals from the donor and emissive signals from the acceptor. - However, to C-13 and C-19 three signals are assigned and two are absorptive. That would not be in line with that idea. Hence, we have to add this case into the increasing list of isotopic labelled samples showing unexpected sign-changes (discussed around line 260). We hope to be able to address this question in future work. Therefore, we added a short statement that the alternating signs are difficult to interpret since the magnetic field strength is close to a turning point (line 290).

7. Page 18: "This confirms that both electron pathways in PSI of duckweed are active and that the electron transfer occurs symmetrically". Symmetrically suggests 50:50 along the two branches. If this is the intended meaning, then I'm not sure where this ratio comes from. Or is "symmetrically" being used rather loosely to mean something like "not exclusively by one branch".

That is a helpful hint. We changed the wording as suggested to "not exclusively by one branch".

Minor comments
Lines 47, 73, 75 : elongates → elongatus? Line 69: PB → $P_B$
Page 18, line 337: extend → extent

Corrected.

---

## Referee Comment (RC2) · Gunnar Jeschke (Referee) · 12 Oct 2020

1. You may want to state that the 2020 Artiukhin *et al.* work uses a frozen density embedding approach for improved treatment of the PS-matrix interaction.

2. I understand that 'bidirectional' was introduced before as a term for what actually is 'two-sided' electron transfer. However, I find this extremely confusing. In all other science and engineering, 'bidirectional' means 'forth and back', which is not what a PS should do under normal condition. It adds to the confusion that in your experiments, with reduced $F_x$, ET becomes bidirectional in the usual sense of the term. There is precedent on using the proper term 'two-sided' (https://doi.org/10.1529/biophysj.105.059824). Please consider using it, too.

3. It is not clear to me how exactly you referenced $^{13}C$ shifts (p. 9, l. 7). Do you quote values with the chemical shift of chlorotrimethylsilane set to 0 ppm or do you set TMS to 0 ppm and assume a known shift for chlorotrimethylsilane? If it is the latter, which shift do you assume? It is about 0.4 ppm difference to normal convention.

4. It is imprecise to state that labeling has an influence on the photo-CIDNP mechanisms. It does have an influence on the outcome, i.e., on the observed nuclear spin polarization.

5. I am not sure about the interpretation in terms of relative contribution of the TSM and DD mechanisms. If isotope labeling changes lifetime(s) of the radical pair, the TSM polarization will also be affected. If it does not, DD should not be suppressed by such labeling. You might want to state that your explanation is tentative.

6. That the polarization of the aliphatic carbon at 52 ppm vanishes at low field implies that polarization transfer by spin diffusion to this carbon is negligible at low field, but not high field. It does not strictly imply that the neighboring aromatic carbons do not obtain enhancement. This should also be formulated with more caution.

7. Conclusion: "Our study contributes to converging and convincing evidence" Please leave it for the readers to decide whether the evidence is convincing. It may be also useful to discuss current limitations. The "which is thought to originate" on page 16 reveals that there is no quantitative understanding (yet) of the supposedly dynamic origin of the asymmetry in bacterial reaction centers. It is also somewhat dangerous to draw conclusions on effects of static electronic structure from only ground-state properties. That chemical shits are similar between the two types of PS in the diamagnetic "resting state" does not necessarily imply that the electronic structure of the donor and acceptor radical states is also similar.

8. Please number pages in the Supplementary Material.

9. Are you sure that a *short* MD calculation would be sufficient to improve chemical shift computations? In other words, can you exclude that chemical shift changes on longer timescales? Rather long MD trajectories would still correspond to the fast chemical exchange limit in NMR. 10. The title does not appear to reflect your main conclusion

Typos/grammar:

p. 2, Line 32, comma after 'properties' is superfluous

p. 3, Line 47: 'Synechococcus elongates' should be typeset italic

p. 4, Lines 67-69: Please be consistent with notation of PA and PB (either always or never subscript)

p. 7, Line 47: superscript missing in '13C'

p. 12, line 60: Please do not jump forth and back between fields and frequencies.

p. 18, line 37: "similar extend" should read "similar extent"

SI, Section 2.5: "calculalted" should read "calculated"

---

## Referee Comment (RC3) · Peter Hore (Referee) · 16 Oct 2020

The authors have dealt satisfactorily with all my comments and suggestions. I am happy to recommend that their manuscript be accepted for publication.

---

## Author Comment (AC2) · 16 Oct 2020

**POINT-BY-POINT REPLY TO REVIEWERS´ COMMENTS**

Analysis of the electronic asymmetry of the primary electron donor of photosystem I of *Spirodela oligorrhiza* by photo-CIDNP solid-state NMR
Janssen, Eschenbach, Kurle, Bode, Neugebauer, de Groot, Matysik, Alia

**REVIEWER GUNNAR JESCHKE**

We thank the reviewer for helpful comments.

1. You may want to state that the 2020 Artiukhin *et al.* work uses a frozen density embedding approach for improved treatment of the PS-matrix interaction.
We added the statement to the reference (page 4, bottom).

2. I understand that 'bidirectional' was introduced before as a term for what actually is 'two-sided' electron transfer. However, I find this extremely confusing. In all other science and engineering, 'bidirectional' means 'forth and back', which is not what a PS should do under normal condition. It adds to the confusion that in your experiments, with reduced $F_x$, ET becomes bidirectional in the usual sense of the term. There is precedent on using the proper term 'two-sided' (https://doi.org/10.1529/biophysj.105.059824). Please consider using it, too.
We thank the reviewer for this helpful advice and changed the wording accordingly (Page 5, lines 87-95 and page 7, line 154).

3. It is not clear to me how exactly you referenced $^{13}C$ shifts (p. 9, l. 7). Do you quote values with the chemical shift of chlorotrimethylsilane set to 0 ppm or do you set TMS to 0 ppm and assume a known shift for chlorotrimethylsilane? If it is the latter, which shift do you assume? It is about 0.4 ppm difference to normal convention.
In fact, we took chlorotrimethylsilane as a reference by mistake, and would like to thank the reviewer for pointing our attention to this issue. We now calculated the $^{13}C$ shielding for TMS with the same methodology, which results in an offset of 9.37 ppm with respect to the value calculated for chlorotrimethylsilane. This is considerably larger than the value of 0.4 ppm mentioned by the reviewer.
But (i) since we speculate that this is more probably due to difficulties in our (non-relativistic) calculation on chlorotrimethylsilane, (ii) it only adds a constant shift to all calculated values presented here, which in no way changes any of the conclusions, and (ii) it actually brings the calculated values overall in better agreement with experiment, we decided to switch to the TMS shielding as a reference. This is now also consistent with experiment.
Therefore, we changed: the Materials & Methods section (page 9, line 207) and in the SI section 2.3 (page 6, first para).

4. It is imprecise to state that labeling has an influence on the photo-CIDNP mechanisms. It does have an influence on the outcome, i.e., on the observed nuclear spin polarization.
That is true. Page 12, line 263: We changed "influence on mechanisms" to "influence on spin-dynamics".

5. I am not sure about the interpretation in terms of relative contribution of the TSM and DD mechanisms. If isotope labeling changes lifetime(s) of the radical pair, the TSM polarization will also be affected. If it does not, DD should not be suppressed by such labeling. You might want to state that your explanation is tentative.
We added a statement limiting the interpretation to samples at natural abundance (p. 12, line 268).

6. That the polarization of the aliphatic carbon at 52 ppm vanishes at low field implies that polarization transfer by spin diffusion to this carbon is negligible at low field, but not high field.

It does not strictly imply that the neighboring aromatic carbons do not obtain enhancement. This should also be formulated with more caution.

Page 12, line 275: We changed "implies" to "might imply" and agree that it is smart to do not rule out other effects.

7. Conclusion: "Our study contributes to converging and convincing evidence" Please leave it for the readers to decide whether the evidence is convincing. It may be also useful to discuss current limitations. The "which is thought to originate" on page 16 reveals that there is no quantitative understanding (yet) of the supposedly dynamic origin of the asymmetry in bacterial reaction centers. It is also somewhat dangerous to draw conclusions on effects of static electronic structure from only ground-state properties. That chemical shifts are similar between the two types of PS in the diamagnetic "resting state" does not necessarily imply that the electronic structure of the donor and acceptor radical states is also similar.

We changed the wording: "Our study contributes to converging and convincing evidence" is now: "Our study suggests". We agree that chemical shift information does not allow to conclude on excited-state properties. DR intensities obtained at the right field might do but we do not want to overstretch the present discussion.

8. Please number pages in the Supplementary Material.

Done.

9. Are you sure that a *short* MD calculation would be sufficient to improve chemical shift computations? In other words, can you exclude that chemical shift changes on longer timescales? Rather long MD trajectories would still correspond to the fast chemical exchange limit in NMR.

The short MD helps with the assessment of the QM optimized structure, because large structural deviations between the MD and QM structures indicate that the structural ensemble is significantly different from the QM single point geometry. We agree that effects happening on longer time scales are not covered by a short MD. A comment on that was added in the manuscript. (page 11 bottom)

10. The title does not appear to reflect your main conclusion

Thank you very much for point this out. We changed the words "electronic asymmetry" to "electronic structure".

Typos/grammar:
p. 2, Line 32, comma after 'properties' is superfluous
p. 3, Line 47: 'Synechococcus elongates' should be typeset italic
p. 4, Lines 67-69: Please be consistent with notation of PA and PB (either always or never subscript)
p. 7, Line 47: superscript missing in '13C'
p. 12, line 60: Please do not jump forth and back between fields and frequencies.
p. 18, line 37: "similar extend" should read "similar extent"
SI, Section 2.5: "calculalted" should read "calculated"

All corrected. Thanks.

---

## Author Response (AR2)

We thank the editor for very careful reading of both, ms and SI.
All typos are corrected.
The good agreement of experimental data between Boxer 1974 and us mentioned.

Replay to the non-public question:
Unfortunately, we were not able to obtain 2D data from that samples allowing us to obtain some solid assignments. Therefore, we cannot assign the two signals from, for example, C-19 to either donor or acceptor sides. One might conclude that the common phase change of the signals at 168.8 and 168.1 (both are assigned to C-19) must occur from either donor or acceptor side (because their field-dependence is identical) and that would strengthen our conclusion that two branches are active. However, we feel the S/N is too poor for such conclusions.